# Metronomic Therapy in Oral Squamous Cell Carcinoma

**DOI:** 10.3390/jcm10132818

**Published:** 2021-06-26

**Authors:** Nai-Wen Su, Yu-Jen Chen

**Affiliations:** 1Department of Internal Medicine, Division of Hematology and Medical Oncology, MacKay Memorial Hospital, No. 92, Sec. 2, Zhongshan N. Rd., Taipei City 10449, Taiwan; medicine_su@hotmail.com; 2Department of Nursing, MacKay Junior College of Medicine, Nursing and Management, Taipei City 112021, Taiwan; 3Department of Radiation Oncology, Mackay Memorial Hospital, No. 45, Minsheng Rd., Tamsui District, New Taipei City 25160, Taiwan; 4Department of Medical Research, China Medical University Hospital, Taichung 40402, Taiwan

**Keywords:** metronomic, oral cancer, oral squamous cell carcinoma

## Abstract

Metronomic therapy is characterized by drug administration in a low-dose, repeated, and regular manner without prolonged drug-free interval. The two main anticancer mechanisms of metronomic therapy are antiangiogenesis and immunomodulation, which have been demonstrated in several delicate in vitro and in vivo experiments. In contrast to the traditional maximum tolerated dose (MTD) dosing of chemotherapy, metronomic therapy possesses comparative efficacy but greatlydecreases the incidence and severity of treatment side-effects. Clinical trials of metronomic anticancer treatment have revealed promising results in a variety cancer types and specific patient populations such as the elderly and pediatric malignancies. Oral cavity squamous cell carcinoma (OCSCC) is an important health issue in many areas around the world. Long-term survival is about 50% in locally advanced disease despite having high-intensity treatment combined surgery, radiotherapy, and chemotherapy. In this article, we review and summarize the essence of metronomic therapy and focus on its applications in OCSCC treatment.

## 1. Introduction

Head and neck squamous cell carcinoma (HNSCC), the sixth most common cancer worldwide, comprises heterogenous groups of tumors arising from the oral cavity, oropharynx, hypopharynx, and larynx [1]. In Taiwan, South–Central Asia, and parts of Europe, oral cavity squamous cell carcinoma (OCSCC) has a higher incidence (9–22 per 100,000) than tumors arising from other sites [2]. Smoking, alcohol consumption, and betel nut use are the major contributors in OCSCC carcinogenesis [3]. More importantly, OCSCC patients exposed to these traditional carcinogens tend to have poorer treatment outcomes than human papillomavirus (HPV)-related HNSCC patients [4]. In addition, the long-term (5 year) survival rate is unsatisfactory (approximately 50%). According to the GLOBOCAN 2018 report, 117,384 of the 354,846 newly diagnosed OCSCC patients died from the disease [5]. Therefore, OCSCC poses a considerable burden to the healthcare systems in different regions [6,7,8,9,10,11]. The majority of OCSCC patients present with a locally advanced disease, and the main therapeutic strategy is tumor wide excision plus radical neck dissection, followed by adjuvant radiotherapy with or without chemotherapy [12,13]. This high-intensity treatment program results in significant acute or chronic adverse events [14]. Locoregional relapse and secondary primary HNSCC [15,16] are the most common recurrent patterns and curable if amenable to local treatment. However, the prognosis is grave once the disease is metastatic or attains platinum-refractory status. Although the therapeutic armamentarium is expanding, for example, anti-epidermal growth factor receptor (EGFR) monoclonal antibody and immune checkpoint inhibitors have been developed, the median overall survival is approximately 10–14 months [17,18,19,20,21].

The most important chemotherapeutic agent in OCSCC treatment is platinum. In platinum-based concurrent chemoradiotherapy or palliative chemotherapy, administration of the maximum tolerated dose (MTD) of cisplatin(100 mg/m^2^ every 3 weeks) is generally considered a gold standard regimen [12,13,17]. The development of platinum-resistant OCSCC is an important factor that leads to treatment failure. Existence of cancer stem cells [22,23], increased angiogenesis [24,25], and unfavorable immune profile in the microenvironment [26,27] are the proposed mechanisms that contribute to platinum-resistant OCSCC and predict poor prognosis. Moreover, the MTD of cisplatin (100 mg/m^2^) is deemed as a highly toxic regimen because it has significant side-effects such as nephrotoxicity, ototoxicity, and neurotoxicity [28,29]. Therefore, novel therapeutic strategies are necessary to improve OCSCC treatment outcomes a step further [30]. Metronomic chemotherapy, one of the potentially promising treatments, is being studied and utilized in clinical practice for a variety of cancer types, including OCSCC [31,32,33,34,35,36,37]. The unique dosing schedule stems from preclinical studies, which have revealed the advantages of overcoming drug resistance and antiangiogenesis effects [38,39]. Clinically, metronomic chemotherapy induces drug sensitivity and, most importantly, it is accompanied by minimal adverse effects [40]. In this article, we briefly review the mechanism of action and clinical development of metronomic therapies in OCSCC.

## 2. History of Metronomic Therapy

Since the appearance of cytotoxic chemotherapeutic agents in the early 1960s, they have been playing a critical role in cancer treatment. The use of MTD to obtain the highest possible cancer cell killing is the mainstream method to calculate the chemotherapy dose [41]. The MTD chemotherapy originated from the great success of a pediatric leukemia treatment model [42]. However, MTD chemotherapy imposes great damage to the rapidly proliferating normal tissues such as hematopoietic or gastrointestinal epithelial cells. To limit the toxic effects and permit patient recovery, MTD chemotherapy should be administered with 3–4 weeks of drug-free interval only. Regrowth of the residual cancer cells is inevitable [43]. Moreover, complete eradication of cancer cells is rarely achieved with MTD chemotherapy owing to the development of drug resistance [44]. Decades of cancer research has revealed that cancer cells possess multiple genetic alterations and could have genomic evolution that contributes to resistance [45,46]. The cancer treatment is far more complex because components of the tumor microenvironment, such as immune cells, endothelial cells, and fibroblasts, communicate and relay pro-survival signals to malignant cells [47,48,49]. This adds more obstacles to our goal of cure or long-term disease control with MTD chemotherapy.

To evaluate the biological effects of administered chemotherapy, other than MTD, one unique dosing schedule—metronomic chemotherapy—was studied in preclinical cell and animal models. Metronomic dosing was defined as administration of low doses (1/3–1/10 of MTD) cytotoxic agents at more frequent intervals (no prolonged drug-free breaks) [50]. Initially, the metronomic chemotherapy was reported to inhibit cancer-associated angiogenesis and consequently promote tumor regression [51,52]. Simultaneously, Broder et al. demonstrated that this low-dose and frequent administration aids in overcoming drug resistance [53]. This regimen is more attractive because it has a lower toxicity profile than MTD [54]. In the following decade, more studies focused on the biological effects of metronomic chemotherapy.

## 3. Metronomic Therapy: Mechanisms of Action

### 3.1. Antiangiogenic Effects

Neoangiogenesis takes a central role during the growing phase of the primary tumor and supports the establishment of distant metastatic deposits. It has long been proposed as a potential target in cancer treatment [55,56]. Chemotherapeutic agents were found to have antiangiogenesis effects in a preclinical study, although the dosing schedule matters in this phenomenon [57]. Tumor-associated vascular endothelial cells are genetically more stable than cancer cells. This is one of the possible explanations why metronomic chemotherapy more specifically causes endothelial cell apoptosis and less resistance development. Other proposed molecular mechanisms are the suppression of endothelial progenitor cell mobilization from the bone marrow [58] and an increase in the expression of thrombospondin-1 (TSP-1), which is an endogenous antiangiogenic factor [59]. Recently, metronomic chemotherapy was found to normalize the defective tumor vasculature [60]. Cyclophosphamide is the prototype of a chemotherapeutic agent possessing antiangiogenic property when administered in the metronomic schedule [61]. The same biological activity was proven for other agents such as taxanes, camptothecin, and vinca alkaloids in a variety of cancer types [62,63]. On the contrary, some studies have suggested that chemotherapeutic agents might induce acute mobilization of the bone marrow-derived circulating endothelial progenitor cells (EPCs), homing viable tumors and promoting their growth. Studies have provided evidence that coadministration of antiangiogenic agents might work in a synergistic fashion with chemotherapy for the suppression of tumor and emergence of resistance [64,65,66,67].

### 3.2. Immunomodulatory Effects

During the process of carcinogenesis and tumor progression, cancer cells may evade the immunosurveillance through the release of cytokines (such as transforming growth factor, vascular endothelial growth factor (VEGF), and interleukin-6) and recruitment of precancerous immune cells (such as regulatory T cell (Treg), myeloid derived suppressor cells (MDSC), and tumor associated macrophage-2) [68]. Studies have shown that the MTD of some traditional chemotherapeutic agents have immunomodulatory effects such as the induction of immunogenic cell death [69]. Through low-dose metronomic chemotherapy, studies found that different agents can deplete Treg and suppress MDSC [70,71,72,73]. In addition, it can promote dendritic cell maturation, an important step to educate the naïve cytotoxic T cells [74]. Some metronomic agents cause cell death and favorably promote the release of “eat me” signals (such as damage-associated molecular patterns, calreticulin, and high-mobility group box 1) [75]. All of these phenomena can possibly activate our immune cells to attack the tumor. However, our immune system is dynamic and, therefore, some study results might be controversial and require further clinical validation [76,77].

### 3.3. Inhibition of Cancer Stem Cells

Cancer cells, like their normal counterparts, contain a minority portion of cells that possess the ability of self-renewal and differentiation. These specific cancer cells are called cancer stem cells (CSCs), which are detected mainly with the expression of specific cell surface markers (CD24^+^, CD44^+^, or CD133^+^; alone or combined expression) [78]. CSCs have been strongly linked to chemoresistance or radio resistance and, consequently, anticancer treatment failure [79,80]. A few studies have revealed that metronomic chemotherapy can decrease the CSC population [81,82,83]. Vives et al. demonstrated that MTD chemotherapy followed by metronomic maintenance therapy with gemcitabine and cyclophosphamide successfully eliminated the CSCs in pancreatic cancer and ovarian cancer orthotopic models, respectively [84]. Incorporation of metronomic chemotherapy into or combined with current cancer treatment may aid in a further reduction in CSC-related resistance.

## 4. Metronomic Therapy in OCSCC

### 4.1. Preclinical Evidence

Although metronomic therapy has been applied in clinical trials or daily oncological practice, only one chemotherapeutic agent, S-1, has been tested completely in both in vitro and in vivo models. S-1 is composed of three compounds. One compound is a 5-fluorouracil (5-FU) prodrug, tegafur. The other two are gimeracil (inhibits the 5-FU degradation) and oteracil (lowers the gastrointestinal toxicity of 5-FU) [85]. In an in vitro study, Ferdous et al. first demonstrated that more frequent 5-FU dosing (16h treatment, 8h rest vs. 96h treatment, 48h rest or 48h treatment, 24h rest) did not result in more cytotoxic cell death according to3-(4, 5-dimethyl-2-thiazolyl)-2, 5-diphenyl-2*H* tetrazolium bromide assay. However, functional evaluations revealed that oral cancer treatment with more frequent 5-FU dosing (16h treatment, 8h rest) showed the highest expression of TSP-1 (antiangiogenesis molecule) and lowest expression of VEGF and CD44 (CSC marker). In the in vivo xenograft model, the usual S-1 schedules were 4weeks on followed by 2 weeks off and 2weeks on followed by 1weekoff, which were the same as the standard regimens in human. The metronomic dosing was 1dayon followed by 1dayoff. The results revealed that all three regimens resulted in similar tumor volume reductions. The nude mice in the two standard regimen groups had a lower weight than those in the metronomic treatment group. The xenograft tumors in the metronomic treatment group showed increased TSP-1 and decreased VEGF expression, which was consistent with the findings of the in vitro study. In addition, a reduction in CD44 expression was observed. The number of endothelial cells (based on CD31 expression) and the micro vessel density (MVD) both decreased in the metronomic treatment group tumors. The results of S-1 treatment in an oral tongue cancer model revealed that metronomic dosing resulted in comparable tumor inhibition. Considering the proof-of-concept viewpoint, metronomic dosing of S-1 demonstrated antiangiogenesis and cancer stem cell elimination effects. According to our knowledge, a variety of compounds extracted from traditional Chinese medicine possess anticancer effects [86]. Some of the studies have tested compounds in xenograft models with daily administration, which mimicked the metronomic dosing schedule. Most studies had a long treatment duration ranging from 3–8 weeks and no significant side-effects [87,88,89,90,91,92,93]. In our previous studies, we treated the oral cancer-bearing mice with cordycepin, a major biological compound from *Cordyceps sinensis*, alone or in combination with radiotherapy for 8 weeks. No weight loss was observed, and the hematological, renal, and hepatic functions were preserved [92,93].

### 4.2. Clinical Evidence

In the past two decades, the concept of metronomic therapy was translated into real practice, and clinical trials have finally proven its effectiveness and better tolerability in a variety of cancers [94,95,96,97] and specific populations [98,99]. Oral chemotherapeutic agents are the central pillars in most studies with different combinations, including targeted therapy [100], vascular disrupting agent [101], and even noncytotoxic agent [102]. We found two main clinical settings for the use of metronomic therapy in OCSCC; one was adjuvant/maintenance therapy after curative intent surgery or chemoradiotherapy, and the other was a palliative therapy in recurrent/metastatic status.

Three studies used metronomic fluorouracil drugs in adjuvant/maintenance therapy. Lin et al. conducted a prospective cohort study and reported that 80 OCSCC patients with stage III–IVa (T3–T4a and/or N0–N1) diseases received curative intent surgery. Forty patients received metronoic UFUR (each capsule contains tegafur 100 mg and uracil 224 mg) with a fixed dose of tegafur 150 mg/m^2^/day for 1 year, and another 40 patients were observational controls. The 4year disease-free survival (DFS) of the metronomic group versus the control group was 85.6% and 75.9% (*p* = 0.02). Simultaneously, they performed laboratory correlation by detecting the change in CEP (circulating endothelial precursor) cells, a marker of angiogenesis activity. The results revealed that patients treated with metronomic UFUR had a significant reduction in the detectable viable CEP cells, which might implicate an antiangiogenesis mechanism behind the survival benefits. The toxicities were mild and manageable, with skin rash (5%) being the most common side-effect, followed by hematological side-effects (2.5%) and nausea/vomiting (2.5%) [103]. Hsieh et al. [104] conducted a retrospective study of 356 stage III and IV OCSCC patients. All the patients received curative surgery followed by chemoradiotherapy, which suggested a more advanced disease condition. A total of 114 patients received tegafur/uracil (tegafur: 100–400 mg/day) as adjuvant therapy for 1–2 years if tolerable. The survival endpoints revealed that 5-year overall survival (OS) and DFS were 65% versus 48% (HR = 0.54, *p* = 0.0008) and 57% versus 41% (HR = 0.62, *p* = 0.0034) in the metronomic and control groups, respectively. In addition, longer UFUR treatment duration (≥12 months) and higher dose (300–400 mg/day) contributed to better survival benefits. The metronomic UFUR treatment seemed to decrease the occurrence of distant metastasis (OR = 4.3, *p* = 0.0015) compared with no UFUR treatment. Moreover, the toxicities were mild and well-tolerated, with skin rash (3.5%) being the most common side-effect. Another study used low-dose S-1 as adjuvant chemotherapy in locally advanced HNSCC patients [105]. The study retrospectively reviewed 52 stage III/IVa/IVb HNSCC patients who completed curative treatment and confirmed no residual tumor. Sixteen out of 52 patients had oral cancer. The S-1 dose was half of the usual standard dose (40 mg/day for body surface area (BSA) < 1.25 m^2^; 50 mg/day for 1.25 m^2^ ≤ BSA < 1.5 m^2^; 60 mg/day for BSA ≥ 1.5 m^2^) and was administered for 2 years. Forty-three (82.7%) patients received S-1 continuously for 2 years without dose reduction. Hematological toxicity was found in 98.1% of the patients; however, only two (3.8%) patients were grade 3. Nonhematological toxicity was found in 15 (28.8%) patients, and all cases were grade 1. The 3year DFS and OS rates were 82.6% and 94%, respectively.

Three studies examined the effectiveness and toxicity profiles using metronomic therapy in recurrent/metastatic OCSCC patients. Patil et al. conducted a prospective randomized phase II clinical trial, comparing metronomic chemotherapy with single agent (three-weekly cisplatin (75 mg/m^2^)) in recurrent/metastatic HNSCC patients. The metronomic chemotherapy consisted of weekly oral methotrexate (MTX; 15 mg/m^2^) and daily celecoxib (200 mg twice daily) [106]. MTX is one of the commonly used chemotherapeutic agents in HNSCC [107]. Celecoxib, a nonsteroidal anti-inflammatory agent, was found to have an antiangiogenesis effect in preclinical experiments [108,109]. A total of 110 patients were enrolled in the study, with 57 randomized to the metronomic arm and 53 to the cisplatin arm. The primary oral cancer patients accounted for 75–77% of the cohort in both the groups. The metronomic arm had a significantly longer progression-free survival (PFS; median PFS: 101 vs. 66 days, *p* = 0.014) and OS (median OS: 249 vs. 152 days, *p* = 0.02) compared with the cisplatin arm. Moreover, the response rate (RR) and clinical beneficial rate were better in the metronomic arm compared with the cisplatin arm: 12.3% versus 1.9% and 54.4% versus 32.1%, respectively. Fewer grade 3–4 adverse events were noted in the metronomic arm compared with the cisplatin arm (18.9% vs. 31.4%, *p* = 0.14), without statistical significance [110]. Due to the promising results of the phase II study, the same group did an open-label, parallel-group, noninferiority, randomized phase 3 trial. The treatments of the experimental arm and control arm were still MTX plus celecoxib versus three-weekly cisplatin (75 mg/m^2^). Overall, 418 patients (211 in the oral metronomic chemotherapy group and 207 in the intravenous cisplatin group) were included in the per-protocol analysis. At a median follow-up of 15.73 months, the primary endpoint, median overall survival, was 7.5 months in the oral metronomic chemotherapy group compared with 6.1 months in the intravenous cisplatin group (HR ratio for death 0.775; 95% confidence interval 0.616–0.974, *p* =0.029). Grade 3 or higher adverse events were observed in 37(19%) of 196 patients in the oral metronomic chemotherapy group versus 61(30%) of 202 patients in the cisplatin group [111]. This was the first phase III study to demonstrate that metronomic chemotherapy resulted in better survival and a preferable toxicity profile in recurrent/metastatic head and neck cancer. Actually, the oral cavity cancers consisted of about 80% of the overall population. The study design and its results undoubtedly added a more solid evidence to the application of metronomic treatment in OCSCC. In another study, a single arm phase I/II trial, Patil et al. determined the dose and evaluated the efficacy of triple metronomic chemotherapy in platinum-refractory oral cancer patients [110]. The regimen included fixed dose erlotinib (an anti-EGFR tyrosine kinase inhibitor; 150 mg/day) and celecoxib (200 mg twice daily). In phase I, they first determined the optimal biological dose (OBD) of MTX by de-escalating from weekly 15 mg/m^2^ and found 9 mg/m^2^ to be the OBD. Overall, 91 patients were enrolled in the study. The median PFS and OS were 4.6 and 7.17 months, respectively. The overall best RR was 42.9%, which was superior to the second line treatment (RR = 2.2–13.6%) in platinum-refractory HNSCC [18,19,107]. The most common grade 3–4 toxicity was hyponatremia (14.8%), followed by elevation of aspartate transaminase (AST) and alanine transaminase (ALT) levels (4.5% and 5.7%, respectively).

## 5. Advantages and Limitations of Metronomic Therapy in OCSCC

Our data review revealed that metronomic therapy does achieve comparable disease control rate in OCSCC and has a favorable toxicity profile compared with standard systemic chemotherapy, such as platinum. However, more solid and high-level evidence (e.g., randomized phase III trials) is required to consolidate the role of metronomic therapy. In addition, we found that the optimal metronomic dosage for OCSCC is not well defined. Pharmacodynamic and pharmacokinetic analysis of the metronomic dosing might provide valuable information for determining the optimal dose [112,113,114]. Considering the viewpoint of healthcare economics, metronomic therapy truly curtails the cost [115]. The reasons for cost reduction are as follows: (1) metronomic agents are not newly developed, (2) enteral administration is performed, and (3) less high-grade toxicities occur that might avoid additional supportive agents or admission. These advantages of metronomic therapy might relieve the soaring healthcare cost and facilitate patients in areas with insufficient medical resources [116]. Although the enteral route for metronomic agents brings convenience and at-home treatment, it should be noted that some OCSCC patients have difficulty in swallowing. This issue matters and affects patient compliance if metronomic chemotherapy is used as adjuvant/maintenance therapy in OCSCC, which might last for 1–2 years [103,104].

## 6. Future Perspectives of Metronomic Therapy in OCSCC

Antiangiogenesis and immunomodulation are the two main mechanisms of action of metronomic therapy [117]. As far as we understand, these two mechanisms might have synergistic effects with contemporary immunotherapy [118,119]. Some preliminary experiments using combined metronomic chemotherapy and vaccine or oncolytic virus in advanced cancers have been reported [120,121]. Although not in OCSCC, some ongoing clinical trials are studying the use of current immune checkpoint inhibitor and metronomic chemotherapy combination in lung cancer (NCT03801304), breast cancer (NCT03007992), pediatric tumors (NCT03585465), and sarcomas (NCT02406781). The results are eagerly awaited, and this strategy can hopefully be tested in OCSCC patients.

## 7. Conclusions

Oral squamous cell carcinoma is a heterogenous disease with a high unmet medical requirement. The cure rate and accompanied complications have not improved despite high-intensity treatment. Metronomic therapy provides an alternative treatment choice in contrast to traditional chemotherapy. Its mechanism of action is as follows: angiogenesis inhibition, immunomodulation, and overcoming drug resistance. A comparative disease control rate and superior toxicity profile are its advantages. It is widely applicable in clinical patient care. However, randomized phase III trials comparing it with the standard care are required to demonstrate its effectiveness. We can anticipate that more studies focusing on metronomic therapy with novel agent combination, such as immunotherapy, will be conducted in the future and translated into clinical treatment benefits.

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
