# Peer review of "Metronomic Therapy in Oral Squamous Cell Carcinoma"

_jcm, 2021, doi:10.3390/jcm10132818_

Round 1
Reviewer 1 Report
Major comment
This is a review article about metronomic therapy in oral squamous cell carcinoma (OCSCC). The topic is interesting due to the scarce availability of effective medical treatments in the advanced stage disease. However, there really are few studies dealing with metronomic chemotherapy in OCSCC published up to now. In fact, this treatment is at the dawn of its use in clinical practice for these patients. This point should be openly stated in the discussion. Indeed, to my knowledge there is just one phase III randomized trial published including OCSCC patients[1], that should be quoted and analyzed in the review.
Minor comments
Please check:
- Line 38: “heath”
- Line 164: “in the”
- Line 254: “…comparable disease control rate” compared with MTD chemotherapy or..?
Acronyms as CEP, which is supposed to be different from EPCs, should be plainly explained.
[1] Low-cost oral metronomic chemotherapy versus intravenous cisplatin in patients with recurrent, metastatic, inoperable head and neck carcinoma: an open-label, parallel-group, non-inferiority, randomised, phase 3 trial.
Patil V, Noronha V, Dhumal SB, Joshi A, Menon N, Bhattacharjee A, Kulkarni S, Ankathi SK, Mahajan A, Sable N, Nawale K, Bhelekar A, Mukadam S, Chandrasekharan A, Das S, Vallathol D, D'Souza H, Kumar A, Agrawal A, Khaddar S, Rathnasamy N, Shenoy R, Kashyap L, Rai RK, Abraham G, Saha S, Majumdar S, Karuvandan N, Simha V, Babu V, Elamarthi P, Rajpurohit A, Kumar KAP, Srikanth A, Ravind R, Banavali S, Prabhash K.
Lancet Glob Health. 2020 Sep;8(9):e1213-e1222. doi: 10.1016/S2214-109X(20)30275-8. PMID: 32827483 Free article. Clinical Trial.
Author Response
Dear reviewer,
Thank you for your time and detailed evaluation of our manuscript.
We made point-to-point response and revision of manuscript according to
your suggestions. Hope this could improve the manuscript a further step.

Reviewer 2 Report
I think this study is well researched. Metronomic therapy is less invasive therapy for cancer. However, the review for the detailed many previous studies less exists. Thus, this manuscript could give the important topic to the oncologists.
I think it is better to include the word ‘review’ in the manuscript title.
The word’ UFUR’ should be closely explained.
In the section ‘4.1. Preclinical Evidence’, it is better to be explained for ‘UFUR’ to Clarify the relationship with S-1.
Author Response
Dear reviewer,
Thank you for your time and detailed evaluation of our manuscript.
We made point-to-point response and revision of the manuscript according
to your suggestions. Hope this would improve the manuscript a step further.
Thank you so much.
Nai-Wen Su, MacKay Memorial Hospital, Taipei.
